# IGF2: A Role in Metastasis and Tumor Evasion from Immune Surveillance?

**DOI:** 10.3390/biomedicines11010229

**Published:** 2023-01-16

**Authors:** Antonino Belfiore, Rosaria Valentina Rapicavoli, Rosario Le Moli, Rosamaria Lappano, Andrea Morrione, Ernestina Marianna De Francesco, Veronica Vella

**Affiliations:** 1Endocrinology Unit, Department of Clinical and Experimental Medicine, University of Catania, Garibaldi-Nesima Hospital, 95122 Catania, Italy; 2Department of Pharmacy, Health and Nutritional Sciences, University of Calabria, 87036 Rende, Italy; 3Sbarro Institute for Cancer Research and Molecular Medicine, Center for Biotechnology, Department of Biology, College of Science and Technology, Temple University, Philadelphia, PA 19122, USA

**Keywords:** insulin-like growth factor 2 (IGF2), isoform A of insulin receptor (IR-A), metastasis, immune evasion, resistance to therapies

## Abstract

Insulin-like growth factor 2 (IGF2) is upregulated in both childhood and adult malignancies. Its overexpression is associated with resistance to chemotherapy and worse prognosis. However, our understanding of its physiological and pathological role is lagging behind what we know about IGF1. Dysregulation of the expression and function of IGF2 receptors, insulin receptor isoform A (IR-A), insulin growth factor receptor 1 (IGF1R), and their downstream signaling effectors drive cancer initiation and progression. The involvement of IGF2 in carcinogenesis depends on its ability to link high energy intake, increase cell proliferation, and suppress apoptosis to cancer risk, and this is likely the key mechanism bridging insulin resistance to cancer. New aspects are emerging regarding the role of IGF2 in promoting cancer metastasis by promoting evasion from immune destruction. This review provides a perspective on IGF2 and an update on recent research findings. Specifically, we focus on studies providing compelling evidence that IGF2 is not only a major factor in primary tumor development, but it also plays a crucial role in cancer spread, immune evasion, and resistance to therapies. Further studies are needed in order to find new therapeutic approaches to target IGF2 action.

## 1. Introduction

Metastasis is the process by which cancer cells escape from primary tumors and migrate to settle in distant organs. Metastatic cancers are usually incurable and are estimated to account for up to 90% of cancer-related mortality [1]. Although recent studies have shed light on some of the mechanisms of metastasis, the molecular players and effectors responsible for the engraftment of tumor cells at distant sites are not well defined. The ability of cancer cells to develop metastasis at secondary sites depends on the capacity of metastatic *foci* to evade immune destruction [2].

Among various factors favoring tumor progression and metastasis, dysregulation of the insulin/IGF system (IIGF) has attracted much attention and is now considered a well-established contributor and possible target [3]. The IIGF comprises three ligands—insulin and insulin-like growth factors 1 and 2 (IGF1 and IGF2). Cognate receptors include the insulin receptor (IR), which comes in two molecular forms—isoform A (IR-A) and isoform B (IR-B)—the IGF1 receptor (IGF1R), and the IGF2 receptor (IGF2R, also known as mannose-6 phosphate receptor, IGFR2/M6FR). Given the high homology between the IR and IGF1R, the two receptors can form heterodimeric hybrid receptors. The ligand binding affinities of the three ligands for the different receptor subtypes are shown in Table 1.

In neoplasia, there are several mechanisms by which dysregulation of the IIGF occurs, including the aberrant expression of IGFs by cancer cells and cells of the cancer microenvironment and the overexpression of cognate receptors [3]. The possible role of peripheral hyperinsulinemia, a common feature of major metabolic disorders, such as obesity and type 2 diabetes mellitus (T2DM), has also received much attention due to its link with cancer [9,10,11,12]. IGFs are produced by cancer cells or by tumor stroma and bind at least six binding proteins with variable affinity [13], while insulin is produced solely by pancreatic beta cells and circulates in the unbound form.

Many studies have addressed the role of IGF1R in tumorigenesis, which has resulted in major efforts to target IGF1R [14]. However, it has become clear that the IR also has an unsuspected role in cancer, as it induces resistance to targeted and conventional therapies, and its overexpression is associated with cell motility, invasion, stem-like phenotype, and increased metabolic flexibility [15]. In fact, the IR is often aberrantly upregulated in many cancers, showing a predominance of IR-A over IR-B, where the IR-A/IR-B ratio almost invariably favors IR-A [3]. IR-A is a dual high-affinity receptor for insulin and IGF2 [5,16,17] and represents the principal receptor for IGF2 in those cancers with IR-A overexpression [7]. However, the role of IGF2 in cancer progression is certainly less characterized when compared to IGF1 and insulin. 

IGF2 is a secreted peptide of 67 amino acids preferentially expressed in early embryonic and fetal development. It is encoded by a 150 kb locus in the region of human chromosome 11, which contains two genes, *Igf2* and *H19*, which are paternally or maternally imprinted, respectively. While *Igf2* encodes IGF2, *H19* transcription generates a long non-coding RNA with a role as a negative regulator of cell proliferation [18]. *H19* might function as a tumor suppressor, but its role in cancer is complex and context-dependent [19]. The *H19* locus also produces miR-675, which inhibits translation of the Igf1 receptor transcript (*Igf1r*), thereby inhibiting IGF2 signaling through the IGF1R [20]. The human *Igf2* gene contains 5 promoters (P1 to P5) and 10 exons, but only the last 3 contain coding sequences, giving rise to different transcripts depending on the specific promoter used [21]. The expressions of transcripts differ in different organs, but in all cases, transcription is driven from promoters P2–P4, with P4 being predominantly active. Transcription of the *Igf2* gene decreases rapidly after birth in most tissues but persists in adult human life, where it is characterized by the biallelic contribution of P1 in the liver [22]. Human P2-derived transcripts are usually expressed at low levels in fetal liver and reach high levels only in transformed cell lines or neoplastic tissues [3]. IGF2 activates the *canonical* IGF signaling pathways by interacting with the IR-A, IGF1R, and IR/IGF1R hybrids [23,24], while IGF2 binding to IGF2R leads to IGF2 degradation into lysosomes or signaling via G proteins [25,26,27,28]. 

IGF2 regulates the metabolism, proliferation, survival, and differentiation of different cell types [29,30,31], and its bioactivity can be enhanced or inhibited by specific IGF-binding proteins (IGFBPs), which show tissue- and age-specific expression [32]. It is worth noting that we previously demonstrated that IGF2 binding to IR-A recruits a somewhat different set of substrates than the binding of insulin to the same receptor [33] and promotes different gene expressions [34]. This difference can be in part explained by the data from our laboratories, which showed that IGF2 was less effective than insulin in promoting the downregulation of IRS1 and IR-A [35], thereby enabling more sustained mitogenic signals. IGF2 is often overexpressed in malignant cells by a variety of mechanisms [36], including re-activation of the fetal *IGF2* promoters, hypermethylation of the *igf2-H19* locus, loss-of-imprinting of the maternal *igf2* allele [18], and reduced expression of the oncosuppressors WT1 and p53 [37,38]. Sustained IGF action promotes cancer development, as shown by transgenic animal models overexpressing IGF2 [39,40]. 

Herein, we focus on evidence that suggests that IGF2 provides a crucial contribution to different hallmarks of the metastatic process, such as stem-like cell phenotypes, the generation of a pre-metastatic niche, and cancer immunoevasion.

## 2. IGF2: A Role in Stem-Like Phenotype of Cancer Cells?

### 2.1. Cancer Cells with Stem-Like Phenotype

Cancer inhibition might require the selective killing of cancer cells with stem-like phenotypes (cancer stem cells—CSCs), which represent a small portion of the cancer cell population, have a pro-metastatic capability, and are particularly resistant to cancer therapies. CSCs principally derive from oncogenic mutations in normal homeostatic stem cells or progenitor cells [41]. Thus, they can proliferate, differentiate, and self-renew like normal stem cells to form various non-CSCs, which make up for the bulk of the tumor [42]. For this reason, they are also called tumor-initiating [43] or tumor-propagating cells [44]. CSCs have developed several mechanisms in order to evade innate and adaptive immunity. Indeed, they possess highly efficient DNA repair and redox tolerance systems, which allow them to sustain immune evasion and resistance to therapy [45]. Due to these unique characteristics, the presence of CSCs in metastatic tumors contributes to cancer-associated death [46]. 

However, the precise identification and targeting of CSCs is a goal not yet achieved. At the same time, there is growing awareness about the fact that cells in the tumor microenvironment (TME) might be particularly supportive of CSCs by exchanging bidirectional paracrine signals essential for maintaining tumor cell stemness [47]. Innovative treatment strategies are therefore focused on targeting these bidirectional signals between CSCs and their niche [48]. 

### 2.2. Autocrine IGF2 and CSCs

Several lines of evidence suggest that autocrine IGF2 plays a role in promoting/maintaining stemness in cancer cells. Recent studies on breast cancer (BC) have revealed a bidirectional regulation between IGF2 and the inhibitor of DNA-binding 1 (Id1) that is linked to cell stemness. Id1 belongs to Id proteins, which bind to basic helix–loop–helix transcription factors and negatively regulate cell differentiation, thereby contributing to maintaining cell stemness [49]. In BC cells, autocrine IGF2 stimulates Id1 expression through the PI3K pathway. In turn, Id1 upregulates IGF2, activating a positive feedback loop to which stem-like BC cells become addicted [50]. In fact, in all BC subtypes, CSCs may become addicted to this IGF2-Id1-IGF2 circuit. Another study demonstrated that Id1 increased the expression and secretion of IGF2 in a variety of human cancer types and that targeting IGF2 inhibited growth and metastasis in mice and enhanced tumor chemosensitivity [51]. Similarly, IGF2 expression was associated with Id1 and activated AKT in esophageal cancer tissues [51]. In addition, Id1-overexpressing esophageal cancer xenografts were characterized by IGF2-dependent metastatic spread [51]. In esophageal cancer cells, IGF2 was crucial in inducing and maintaining stemness through the PI3K/Akt/MiR-377/CD133 pathway [52]. Blockade of IGF2 by a neutralizing antibody inhibited stem-like cell features and suppressed the growth of esophageal cancer xenografts.

Other oncogenic circuits can also exploit IGF2 production to sustain cancer cell stemness. For instance, the CD74-Neuregulin1 (*NRG1*) fusion gene, a driver of invasive mucinous lung adenocarcinoma, activates the HER2/HER3/PI3K/NF-κB pathway, also leading to autocrine IGF2 production and CSC maintenance [53].

In this context, it is interesting to note that Id1 attenuates interferon regulatory factor 3 (IRF3) promoter suppression induced by the transcription factor FOXO1 [54], derepressing IFN signaling. According to these findings, we can speculate that the activation of the IGF2-Id1-IGF2 loop stimulates chronic aberrant IFN signaling in cancer cells, which contributes to stem-like phenotype and chemoresistance [55]. However, further studies are needed to demonstrate the exact relevance of inhibiting FOXO1 transcriptional activity in promoting the stem-like phenotype of IGF2 in cancer (Figure 1).

Notably, transcription factors implicated in promoting and maintaining cell stemness also stimulate the autocrine production of IGF2 by cancer cells. Among these transcription factors, SOX2 epigenetically upregulates IGF2 expression, as revealed by a strong H3K4me3 signal, a marker of active promoters, at the IGF2 locus of SOX2-expressing cells. IGF2 in turn activates the IGF1R/Akt pathway and enhances aggressiveness, stem-like features, and overall survival of bladder cancer [56] (Figure 1). SOX2 is a stemness-related transcription factor essential for self-renewal, and, together with OCT4, MYC, and KLF4, it is important for reprogramming differentiated cells into induced pluripotent stem cells (iPSC) [57]. SOX2 expression is also implicated in several cancers, including breast, esophagus, and lung [58,59]. IGF2 is a known target of miRNAs, such as miR-100 and miR-125b in embryonic and cancer cells [60,61]. In hepatocarcinoma cells, loss of miR-100 and miR-125b enhances stem cell features through the upregulation of the IGF2/Akt/mTOR pathway [62] (Figure 1). 

### 2.3. Paracrine IGF2 and CSCs

Recent studies have implicated paracrine IGF2 in the induction and maintenance of epithelial–mesenchymal transition (EMT) and stem-like phenotype of cancer cells [63,64]. Cancer-associated fibroblasts (CAFs) represent a heterogenous component of the TME and play a key role in the regulation of various aspects of cancer biology, including angiogenesis, metabolic reprogramming, metastasis, and immune evasion [65,66]. As early as 1995, it was reported that fibroblasts from normal breast predominantly secreted IGF1, while CAFs from invasive tumors preferentially secreted IGF2. Other soluble mediators released by epithelial tumor cells, such as β fibroblast growth factor (βFGF) and transforming growth factor β (TGFβ), contribute to the activation of fibroblasts, which in turn secrete IGF2 [67,68] (Figure 2). 

According to the results of Chen et al., CAFs originating from primary lung tumors, but not normal fibroblasts, supported lung CSCs cultured as spheroids and allowed for long-term CSC cultures, which maintained elevated levels of Nanog, Sox2, and Oct3/4 [69]. The removal of CAFs reduced the expressions of stemness markers in lung CSCs, which showed a tendency to differentiate and a reduced ability to metastasize [69]. These findings resemble the ability of embryonic fibroblasts to support human embryonic stem cells [70]. Transcriptional analysis showed that CAFs expressed high levels of growth factors, including IGF2, hepatocyte growth factor (HGF), LIF, and CXCL12, while lung CSCs expressed their cognate receptors. Interestingly, IGF2 was the factor predominantly involved in maintaining cancer cell stemness through a paracrine mechanism involving the IGF1R/Nanog pathway. Accordingly, Nanog silencing abrogated the effect of IGF2, while IGF1R blockade by a specific antibody or a TK inhibitor inhibited Nanog expression and tumorigenesis. Notably, in 80 lung cancer samples, the IGF2/IGF1R paracrine loop significantly correlated with Nanog expression and poor overall survival. Finally, the study provided evidence that IGF2 expression in CAFs promoted a variety of CAF-secreted cytokines/growth factors, including granulocyte–macrophage colony-stimulating factor, βFGF, HGF, and IGFBP2 (Figure 2). Consistent with these findings, in colon cancer, IGF2 produced specifically by CAFs induced the myofibroblast differentiation of these cells and mediated physical matrix remodeling favoring tumor invasiveness and dissemination. Moreover, the expression of IGF2 in colon cancer was associated with poor prognosis [71]. In addition to CAFs, other cells of the TME produced significant amounts of IGFs, especially IGF2, such as M2-like tumor-associated macrophages (TAMs). In the anaplastic thyroid cancer model, IGF1 and IGF2 produced by M2-like TAMs promoted invasion and stem-like features in epithelial cancer cells by binding to IR-A and IGF1R and activating the PI3K/AKT pathway [64] (Figure 2).

## 3. IGF2 Role in the Pre-Metastatic Niche and Metastasis

The intrinsic properties of tumor cells and a cancer-favoring environment at distant sites are essential for the initiation of metastasis. This environment is set long before tumor metastasis occurs and is called the “pre-metastatic niche”, whose formation includes the reprogramming of resident cells, myeloid cell recruitment, and matrix remodeling [72], thereby providing a fertile soil for circulating tumor cells (CTCs) that have entered blood vessels. Compared to orthotopic cancer cells, CTCs have a unique phenotype and gene expression [73], enabling them to metastasize at distant sites by adapting to the dynamic environment of the bloodstream and interacting with various cellular components.

Interestingly, tumor-derived IGF2 has been implicated not only in educating cells of the primary tumor microenvironment but also in shaping the metastatic macroenvironment by stimulating the formation of pre-metastatic niches. For instance, in esophageal cancer models, IGF2 secreted by Id1-overexpressing cells induced CAFs to produce and secrete vascular endothelial growth factor (VEGF) by inhibiting miR-29c [74]. In turn, VEGF mobilized VEGFR1+ve bone marrow cells (BMDCs) to premetastatic sites in the lung. In this context, the chemokine CXCL5 produced by lung cells also plays a role in favoring lung colonization by esophageal cancer cells that express the cognate receptor CXCR2 [74].

Gui et al., investigated whether the gene expressions of CAFs from primary breast tumors (pCAFs) were different from CAFs obtained from various metastatic sites (mCAFs). By comparing the transcriptomes of pCAFs and mCAFs, they discovered some significant differences, including a significant IGF2 upregulation in mCAFs [75]. IFN-related genes were predominantly expressed in non-liver mCAFs. Indeed, mCAFs supported tumor growth and metastasis in orthotopic mouse xenografts of BC cells and induced mesenchymal-like changes in BC cells when co-cultured in vitro. The critical importance of IGF2 was confirmed by the observation that treatment with an anti-IGFs blocking antibody inhibited in vivo tumor growth derived from a coculture of BC cells and mCAFs injected into mice. The authors also demonstrated that mCAFs had more potent immunosuppressive action on T effector cells than pCAFs and fibroblasts from normal breast. These effects could be partially blocked by treatment with anti-IGF blocking antibodies.

IGF2 promoted brain metastases in triple-negative breast cancers (TNBC). Indeed, brain vessel pericytes secrete high levels of IGF2 (>1000 pg/mL-1), which stimulates proliferation and maintains the stem-like features of metastatic TNBC cells [76]. In cholangiocarcinoma, a desmoplastic cancer of the biliary tree, treatment with epidermal growth factor receptor (EGFR) inhibitor erlotinib induced the generation of cells with mesenchymal properties and stem-like phenotypes with the upregulation of both IR and IGF1R and concomitantly increased the expression of unprocessed forms of IGF2, which have a similar binding affinity but higher bioavailability than mature IGF2. These tumors showed enhanced proliferation of CAFs, which determined the IGF2-evoked proliferation of both CAFs and tumor cells [63] (Figure 3). The authors did not fully characterize the receptors mediating the IGF2-dependent responses, but both IGF1R and IR (especially the IR-A isoform) were expressed and activated.

## 4. IGF2 and Immunosuppression

The immunosuppressive network includes myeloid-derived suppressor cells (MDSCs), regulatory T cells (Tregs), regulatory B cells (Bregs), M2 macrophages (Mreg), regulatory subsets of dendritic cells (DCreg), NK cells (NKreg), and natural killer type II (NKT) T cells, which show regulatory and immunosuppressive activities [77]. In this cooperative network, the enhancement of immunosuppressive properties and suppression of pro-inflammatory immune effector cell functions occur through the secretion of anti-inflammatory cytokines, such as IL-10, TGF-β, IL-4, IL-27, and IL-35, and the release of reactive oxygen and nitrogen species (ROS/RNS). Immunosuppressive cells act through several mechanisms, two of which are the expressions of arginase 1 (ARG1) and indoleamine 2,3-dioxygenase (IDO), which reduces L-arginine and tryptophan in the TME, thereby suppressing immune effector cell function [77]. Furthermore, immunosuppressive cells increase the expressions of immune checkpoint proteins, such as programmed cell death protein 1/programmed death-ligand 1 (PD-1/PD-L1) and CTLA-4 proteins, which suppress T cell activation by preventing antigen presentation (Figure 4).

There is evidence that IGFs, namely IGF2, play a role in tumor immunoevasion by regulating the complex immunosuppressive network at multiple levels.

### 4.1. IGF2 and Dendritic Cells (DCs)

DCs link innate and adaptive immune responses and are key regulators of T and B cell immunity for their ability to uptake, process, and present antigens. It has been shown that IGFs can initiate DC maturation and inhibit their apoptosis [78], allowing them to secrete different cytokines, such as IL-10. These cytokines can improve the immunosuppressive status of the TME and suppress tumor-associated antigen-specific T cell immunity [79,80]. Furthermore, elevated levels of IL-10 and IL-6 can promote a Th2 inhibitory immune response [81]. On the other hand, Somri-Gannam et al., demonstrated that IGFs suppressed DC maturation [82]. Indeed, IGF-treated DCs showed a decreased antigen processing capacity, indicating that they can act as tolerogenic DCs [82]. In this model, the IGF1R tyrosine kinase inhibitor AEW541 blocked the suppressive effects of IGFs on DCs, thereby proposing IGFs as novel targets to generate potent antitumor immunity by rescuing DCs’ impaired function.

### 4.2. IGF2 and Myeloid-Derived Suppressor Cells

Myeloid-derived suppressor cells (MDSCs) include a population of immature myeloid cells with an important immunosuppressive role as, in fact, they suppress T cell activity [83]. In the TME, this cells population is expanded in response to various tumor-derived factors, including IL-4, IL-6, IL-13, and TGFβ, which impair the maturation of bone marrow-derived myeloid cells (BMDCs) into antigen-presenting cells (APCs) [84]. Thus, the expansion of MDSCs contributes to tumor immune evasion by reducing the maturation of APCs, especially DCs, and producing diverse mediators, such as arginase, reactive oxygen species, and nitric oxide, which suppress the activity of T cells. 

Papaspyridonos et al., demonstrated that Id1 overexpression induced MDSC accumulation and DC/MDSC imbalance, leading to an immunosuppressive phenotype and increased tumor growth [85]. Id1 overexpression was associated with an increased level of ROS, which modulated MDSCs expansion. In their melanoma model, the authors identified TGFβ and IL-6 among the top predicted upstream regulators of Id1 overexpression in MDSCs. However, these factors are context-dependent and, given the well-known relationship between Id1 and IGF2, it could be hypothesized that IGF2 has a similar role in other contexts (Figure 4).

Notably, a recent study identified PI3K in a systematic screen for modulators of resistance to immunotherapy [86]. In this model, the activation of PI3K signaling in cancer cells was characterized by the CCL2-dependent recruitment of inhibitory CD45+ CD11b+ Ccr2hi myeloid cells, a reduced infiltration of CD8+ T cells after immunotherapy, and resistance to immune checkpoints blockade. While this study was conducted in a cancer model with activating mutations of PI3K, it is likely that other mechanisms may contribute to PI3K activity, such as PTEN mutations or IR/IGF1R activation by autocrine IGF2. Overall, these findings do not provide evidence of direct involvement of IGF2 in the expansion of MDSCs but demonstrate that MDSC expansion is closely related to Id1 expression and constitutive PI3K activity, both of which are often dependent on IGF2 action.

### 4.3. IGF2 and T Cells

T cells play a central role in anti-tumor immunity, as they exert vital roles in tracking and killing cancer cells in tumors. After recognizing tumor antigens, T cells are activated and amplified to exert anti-tumor effects. However, cytotoxic T cells of the TME commonly show a state of exhaustion associated with elevated expressions of PD-1 and CTLA-4 [87,88] and other dysfunctions [89] related to the tumor infiltration of inflammatory-CAFs (iCAFs). Wu et al. [90] carried out single-cell RNA sequencing of TNBC samples and isolated two different subpopulations of CAFs, categorized as myofibroblasts (myoCAFs) and iCAFs, the latter being characterized by high expressions of growth factors, including IGF1, IGF2, FGF, and PDGFD, and chemokines, such as CXCL12 and CXCL13. A third important stromal population was represented by cells with a perivascular-like (PVL) profile, otherwise called “vascular-like CAFs”, which were not necessarily associated with the endothelium. They demonstrated that gene signatures from iCAFs and differentiated PVL cells were strongly associated with cytotoxic T cell dysfunction. However, the authors did not investigate the role of IGF2 or other specific factors in determining T cell dysfunction. Although research in the field is still preliminary, several lines of evidence support a connection between IIGFs and the expansion and increased activity of Treg cells and immunosuppression. Tregs are characterized by epigenetic regulation by the FoxP3 transcription factor, which induces Treg differentiation and the activation of immunosuppression [91]. Tregs play a key role in maintaining self-tolerance and immune homeostasis by negatively controlling effector T cells and cells of innate immunity, such as natural killer (NK) cells [92]. Mechanistically, IIGFs-evoked STAT3 activation increases Treg maturation by enhancing FoxP3 transcription [93]. IGF1 specifically stimulates the in vitro proliferation of Tregs by increasing FoxP3 expression [94].

Using a model of allergic inflammation of the intestine, Yang et al. [95] showed that IGF2 stimulated the proliferation of Treg cells, likely via the IGF2R/MAPK pathway, as in fact, it was sensitive to IGF2R silencing. In the presence of concomitant TCR stimulation, IGF2 boosted Treg cell function, as shown by TGFβ secretion. This effect was attributed to IGF2 binding to IGF2R, although not mechanistically demonstrated by gene depletion. Indeed, Treg cells expressed IGF2R and low levels of IGF1R. However, the IR-A was not taken into consideration, and therefore, its role in mediating IGF2 action cannot be excluded. On the other hand, IGF1 stimulated the in vitro proliferation of human and mouse Treg cells and inhibited disease progression by binding the IGF1R when injected in vivo into mouse models of autoimmune disease [94]. IGF1 stabilized the Treg transcriptional signature and enhanced FoxP3 expression. These findings are in agreement with studies showing that mesenchymal stem cells (MSCs) exert anti-inflammatory effects in part by secreting IGF1. Indeed, both IGF1 and IGF2 in hMSC culture supernatant potentiated the induction of CD4+FOXP3+ Tregs. This effect was inhibited by IGF1R silencing. In contrast, IGF2R silencing enhanced the effect of IGF2 likely by increasing the bioavailability of IGF2 [96]. Other studies have reported the specific effects of IGF1/IGF1R by favoring the expansion of Tregs but not of other T cells subtypes [94,97]. Overall, these studies have indicated that IGF2 may favor Treg expansion and immunosuppressive activity. Although some studies have implicated IGF2R in this response, IGF1R is also likely involved. However, more investigations are needed to fully address the effects of IGF2 in Treg expansion in tumors and the differential role of different IGF2 receptors, including the IR-A, in mediating Treg modulation.

### 4.4. IGF2 and Macrophages

The macrophages of the TME are called tumor-associated macrophages (TAMs), and they play an important role as the primary immune cells within the TME. Tissue-specific resident macrophages and newly engaged monocytes are recruited to the TME and differentiate into TAMs under the control of growth factors and chemokines produced by tumor and tumor stromal cells [98]. TAMs are polarized into two phenotypes, M1 (classically activated) and M2 (alternatively activated) TAMs, which play different roles in TME. The recruitment and polarization of TAMs are orchestrated by tumor- and host-derived cytokines and chemokines [98]. M2 polarization of macrophages usually requires IL-4 and IL-13 stimulation. However, a subset of TAMs can acquire an M2-like phenotype in response to lactic acid activating HIF1a and independently of IL-4Ra signaling [99]. M2 TAMs are characterized by increased glycolysis, especially in the early inflammatory phase of cancer onset [100]. These findings raise the possibility that IIGF signaling may favor M2 polarization of TAMs by enhancing tumor glycolysis.

Interestingly, metabolic changes and IIGF signaling play pivotal roles in determining macrophages’ polarity and their pro-inflammatory or anti-inflammatory action. During activation by LPS, macrophages shift to aerobic glycolysis and reduce the oxidative phosphorylation (OXPHOS) pathway [101]. However, IL-4-induced M2 macrophages showed an increase in both glycolysis and OXPHOS [102]. Macrophages express all IIGF receptors (IR, IGF1R, and IGF2R) and respond functionally to IGF2, IGF1, and insulin. The functional role of IGF2 in TAMs is still unclear, although some clues come from the study of non-neoplastic inflammatory conditions. In the setting of experimental autoimmune encephalomyelitis, Du L et al., showed that IGF2 produced by hypoxic mesenchymal stem and/or stromal cells (MSCs) can significantly alter the maturation of macrophages by committing them for OXPHOS even in the context of inflammation [103]. IGF2-primed macrophages showed high levels of programmed death-ligand 1 (PD-L1) expression and a low expression of IL-1β and enhanced the functionality of Tregs through PD-L1-PD-1 interactions. In a follow-up study performed in a model of DSS-induced colitis, the same authors demonstrated that IGF2 can play a dual role in macrophages. Macrophages exposed to low doses of IGF2 (≤50 ng per mouse) indeed favored OXPHOS and acquired an anti-inflammatory phenotype, showing low levels of IL-1β and high levels of PD-L1 expression. In contrast, macrophages exposed to high doses of IGF2 (1000 ng per mouse) were instead committed to glycolysis and acquired a pro-inflammatory phenotype [104]. This dual role was associated with IGF2 binding to different receptors. At low doses, IGF2 binds the high-affinity receptor IGF2R, which internalizes and translocates into the nucleus, thereby inducing GSK3-mediated methylation at the *v-ATPase* gene promoters with subsequent proton rechanneling in the mitochondria. At high IGF2 doses, instead, the action is mediated by the IGF1R, which has a lower affinity for IGF2 than the IGF2R, and induces DNA demethylation, which counteracts the effects of IGF2R and induces a pro-inflammatory phenotype. Remarkably, another study reported that IGF1 may stimulate trained immunity in monocytes via the IGF1R/mTOR pathway, which is under the control of metabolic factors, such as mevalonate, along the cholesterol biosynthetic pathway [105]. According to this study, glycolysis-induced mevalonate synthesis increased IGF1R signaling, which in turn further increased glycolysis. Trained immunity is a form of non-specific memory ensuring an enhanced function of the innate immune system [105], which requires epigenetic changes via histone modifications. These findings suggest, therefore, that IGF1 and IGF2 can activate different programs in monocytes. 

In a model of THP-1, macrophages challenged with oxidized low-density lipoprotein (ox-LDL), IGF2, NF-κB, and IL-6 were upregulated, and IGF2 stimulated IL6 secretion via NF-κB [106]. These studies have not considered the possible effect of IGF2 binding to the IR-A isoform, which binds IGF2 with a similar affinity as the IGF1R. In fact, when both IGF1 and IGF2 are co-expressed, IGF2 is expected to preferentially bind the IR-A rather than the IGF1R, as the latter predominantly binds the high-affinity ligand IGF1. However, in vivo studies have been performed in mouse models of chronic inflammation and not in cancer models. Thus, further research effort is needed to demonstrate whether high IGF2 may similarly program macrophages of the tumor environment. The effect of insulin on macrophages is well studied, especially in the context of insulin resistance in obesity and diabetes [107,108]. Notably, the macrophages of animals fed a high-fat diet were insulin-resistant, acquired an M2-like phenotype, and had reduced LPS responses. They showed increased basal mTORC1 activity and increased basal glycolysis [107] but lacked Akt2 activation in response to insulin. This phenotype was recapitulated by Akt2 silencing, suggesting that M2 polarization is sustained by Akt1 but suppressed by Akt2 activation. Indeed, another study showed that the deletion of Akt1 promoted M1 polarization [109]. These studies, although not conducted in TAMs, have shown that the pro-inflammatory environment elicits insulin resistance in macrophages. In these conditions, chronic stimulation of the IR can favor M2 polarization in macrophages. 

### 4.5. IGF2 and Immune Checkpoint Molecules

PD-L1 is a transmembrane protein expressed primarily on cancer cells and binds to its receptor, PD-1, expressed by activated T and B cells. The interaction of PD-L1 with PD-1 inhibits T lymphocyte proliferation, cytokine production, and cytolytic activity [110,111], which induces the functional exhaustion of T cells and suppresses their anti-tumor immune responses [87]. Accordingly, antibody-mediated inhibition of the PD-1/PD-L1 pathway reactivates T cell function against tumor cells [112]. Indeed, this immunotherapy strategy has given promising clinical results in several cancers, including melanoma, renal cell carcinoma, and lung cancer, thereby opening new therapeutic options for these neoplasia [113].

Tumor cells mostly express PD-L1 following stimulation by interferons or other inflammatory cytokines [114]. However, cancer cells can also present constitutive PD-L1 expression as a consequence of multiple mechanisms driven by oncogene mutations or the loss/inactivation of tumor suppressors (e.g., PTEN mutations), leading to the activation of key intracellular cascades, such as the JAK/STAT3, [115] the PI3K/AKT/mTOR, or the ERK1/2 pathways [44,116].

Notably, PD-L1 expression usually increases in CSCs [117]. In colon cells cultured as spheroid and enriched in CSCs, insulin increased PD-L1 expression and transport to the cell membrane through the PI3K/Akt pathway [118], thereby suggesting that PD-L1 expression in colon CSCs is dynamic and responsive to insulin in the tumor environment. However, the study did not evaluate the effect of IGF2 and the cognate receptors involved. In addition, the authors used relatively high doses of insulin, which can stimulate both IR and IGF1R. Studies in pancreatic ductal adenocarcinoma (PDAC) showed that not only insulin but also IGF1 and IGF2 upregulated PD-L1 expression through the activation of the ERK pathway. Notably, this response was mediated by both the IGF1R and the IR-A isoform [119] as, in fact, high levels of PD-L1 induction by insulin were observed in cells highly expressing IR-A and IGF1R. Insulin directly stimulated PD-L1 transcription by activating its promoter activity. Moreover, the IR and PD-L1 colocalized in the cytoplasm of cancer cells. As a consequence of this interaction, PDAC cells exposed to insulin suppressed the proliferation of activated human CD8+ T-cells, which was reversed by either PD-1 or PD-L1 blockade. Taken together, these studies suggest that peripheral hyperinsulinemia but also paracrine/autocrine IGF2 can favor immune evasion in certain cancers by upregulating the expression of immune checkpoint PD-L1.

Interestingly, several lines of evidence indicate that PD-L1 can also elicit non-immune oncogenic effects in cancer cells. For instance, PD-L1 promoted EMT in TNBC cells by stabilizing Snail through the inhibition of the glycogen synthase kinase 3β (GSK3β), whose activation stimulates Snail ubiquitination and degradation [120]. In fact, in a variety of cancer types, PD-L1 expression and EMT status are linked by a bidirectional relationship [121,122,123]. PD-L1 expression in tumor cells also stimulated glycolysis [124,125] and lipid uptake, thereby favoring a selective advantage of malignant cells over tissue-resident memory T (Trm) cells, resulting in immunosuppression [126].

## 5. Therapeutical Implications

Several studies have shown that both autocrine and paracrine IGF2 are associated with drug resistance in various cancer models and that targeting the IGF signaling pathway can sensitize cancer cells to therapy. 

Autocrine IGF2 induced resistance to taxol in BC cells [127], while in non-small-cell lung cancer cells, CAF-dependent activation of the IGF2-AKT-Sox2-ABCB1 signaling was associated with resistance to doxorubicin, which was reverted by IGF1R blockade [128]. Moreover, in osteosarcoma cells, chronic exposure to IGF2 induced a dormancy-like state, which was associated with resistance to various chemotherapeutic drugs [129]. This effect was replicated by insulin, suggesting a putative role of the IR-A. In HER2-positive breast cancers, resistance to the anti-HER2 blocking antibody Herceptin is associated with tumor recurrence and distant metastases. Interestingly, these cells show aberrant IGF2/IGF-1R/IRS1 signaling, consequent to disruption of the FOXO3a-miRNA negative feedback inhibition, which plays a role in Herceptin resistance [130]. In pancreatic cancer, macrophages and myofibroblasts are the main sources of IGF2 and IGF1, which trigger stimulatory effects via IR-A and IGF1R. This tumor-promoting action was reverted by IGF-blocking antibodies [131]. 

Inflammatory/immune mechanisms can promote drug resistance and the prometastatic effects of IGF2. In hepatoma cells, IGF2 treatment resulted in increased inflammation, which repressed DNA repair enzyme expression and caused DNA damage [132]. Autocrine IGF2 is also implicated in cancer resistance to IGF1R blockade [133,134,135]. In this regard, Lee J-S et al., demonstrated that mice with orthotopic breast tumor transplants treated with the anti-IGF1R monoclonal antibody cixutumumab presented increased metastasis and decreased overall survival. Mechanistically, IGF1R blockade stimulated a STAT3-dependent increase in autocrine IGF2 production by cancer cells, which recruited macrophages and fibroblasts, leading to the production of CXCL8 and proangiogenetic and prometastatic actions [136]. Significantly, these effects were abolished by silencing IGF2R expression in macrophages and fibroblasts or IGF2 in cancer cells, indicating that the IGF2/IGF2R interaction played a pivotal role in mediating these effects and suggesting that IGF2 or STAT3 could represent suitable co-targets in combination with IGF1R blockade. In agreement with these studies, Hashimoto et al., showed that the efficacy of immunotherapy with anti-PD-1 antibodies in reducing the growth of experimental PDAC liver metastases was enhanced by blocking IGF2 and IGF1 signaling using an IGF-TRAP, a fusion protein entailing the extracellular domain of the human IGF1R fused to the Fc portion of human IgG1 [137]. Interestingly, this treatment partially reverted the immunosuppressive landscape associated with PDAC liver metastases.

Taken together, these studies strongly suggest that targeting IGF2 signaling in combination with chemotherapy/targeted therapy as well as immunotherapy could be a viable approach for therapy.

## 6. Conclusions

The biological role of IGF2 in prenatal growth and development is widely recognized. However, its physiological role in adult life is not very well characterized in comparison to the IGF1 homolog. The same is true when considering the role of IIGFs in cancer. Here, we covered studies that have provided compelling evidence that IGF2 is not only a major factor in primary tumor development, but it also plays a crucial role in cancer spread, immunoevasion, and resistance to therapy. Of note, IGF2 is approximately five-fold more abundant than IGF1 in serum and is often more expressed than IGF1 in cancer, in both an autocrine and paracrine manner. Compared to IGF1, IGF2 binds the IGF1R with a lower affinity but binds the IR-A with a much higher affinity. Notably, IR-A is often preferentially overexpressed in advanced cancer compared to IGF1R. Particularly intriguing is the evidence suggesting that IGF2 expression can promote cancer immunoevasion, which is therefore another response elicited by IGF2, in addition to tumor cell motility, invasion, stem phenotype, and angiogenesis, thereby indicating that IGF2 is a pleiotropic promoter of metastatic spread in cancer. However, the specific contribution of different IGF2 receptors is not well defined and needs further characterization. Overall, the body of these studies provides a strong rationale for exploring therapeutic strategies for advanced cancers, where IGF2 blockade could be used in combination with other approaches, including immunotherapy.

## Figures and Tables

**Figure 1 biomedicines-11-00229-f001:**
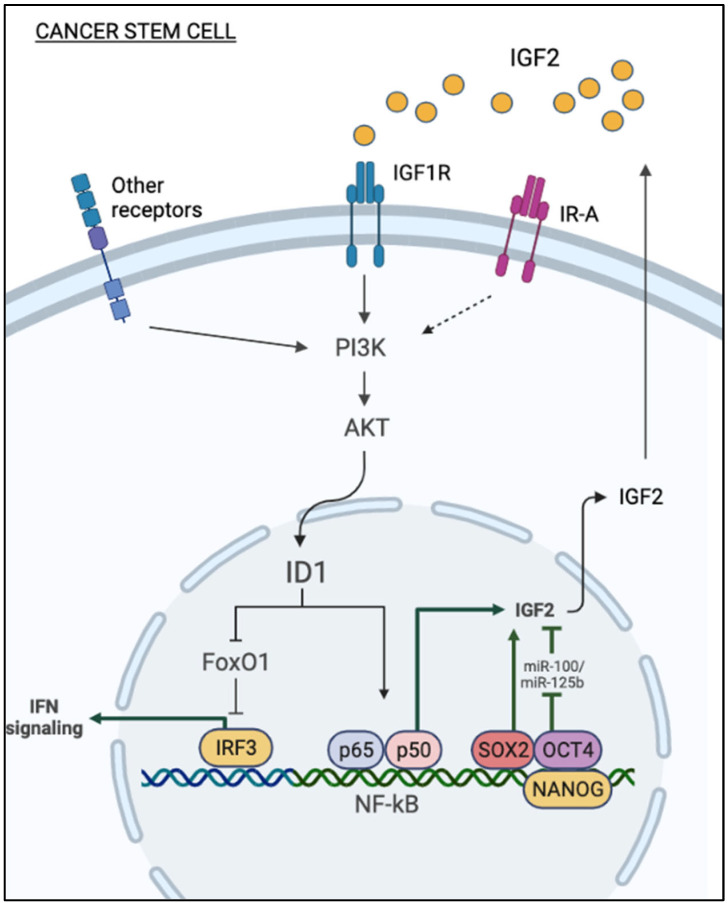
Mechanisms of autocrine production of IGF2 in cancer stem cells. Autocrine IGF2 stimulates Id1 expression through the PI3K/AKT pathway. Id1 relieves IRF3 promoter suppression induced by the transcription factor FOXO, thereby derepressing IFN signaling. In turn, Id1 upregulates IGF2, activating the NF-κB pathway. Transcription factors implicated in cancer cell stemness (i.e., SOX2, OCT4, Nanog) can play a role in stimulating the autocrine production of IGF2. Loss of expression of miR-100 and miR-125b enhances stem cell features of cancer cells through the upregulation of the IGF2/Akt/mTOR pathway.

**Figure 2 biomedicines-11-00229-f002:**
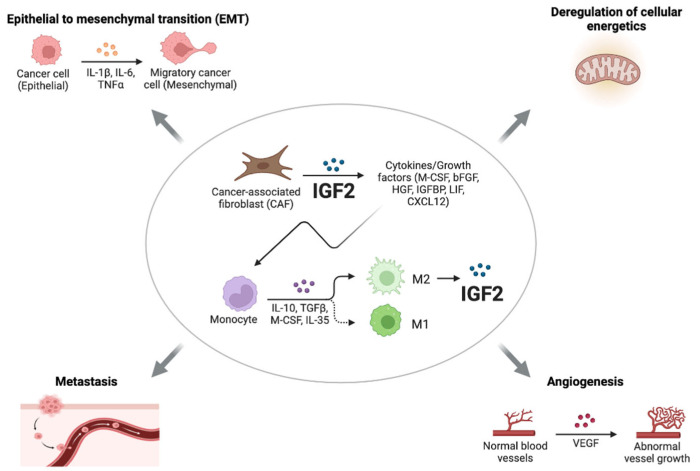
Roles of paracrine IGF2. CAFs from invasive tumors preferentially secrete IGF2, which induces the production of a variety of cytokines and growth factors. In addition to CAFs, other cells of the TME, such as M2-like tumor-associated macrophages (TAMs), secrete high levels of IGF2, thereby promoting invasion and stem-like features in epithelial cancer cells through the IR-A and IGF1R and downstream activation of the PI3K/AKT pathway. IGF2 produced by CAFs can also play a key role in the regulation of various aspects of cancer biology, such as angiogenesis, metastasis, EMT, and the deregulation of cellular energetics.

**Figure 3 biomedicines-11-00229-f003:**
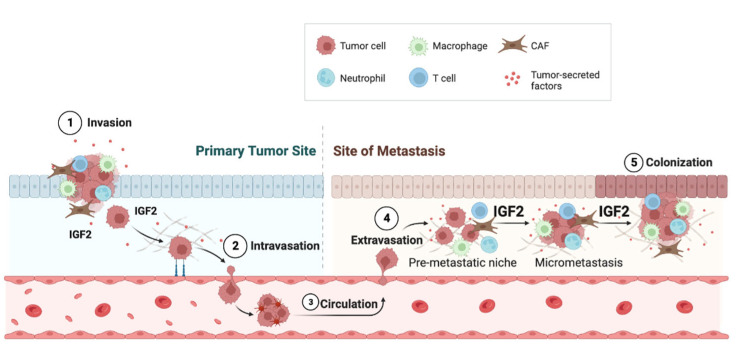
IGF2 promotes metastasis. Tumor-derived IGF2 affects cells of the microenvironment but also promotes distant metastatic spread by stimulating the formation of pre-metastatic niches. Compared to CAFs derived from primary tumors (pCAFs), metastatic CAFs (mCAFs) produce high levels of IGF2, which supports tumor growth and metastasis. In addition, IFN-related genes are predominantly expressed in mCAFs.

**Figure 4 biomedicines-11-00229-f004:**
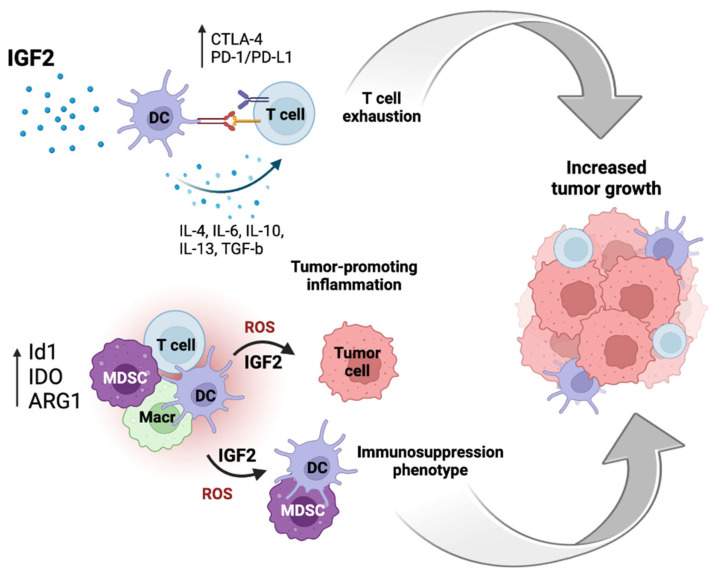
IGF2 plays a role in anti-tumor immunity suppression in the tumor microenvironment (TME). IGF2 induces the maturation of dendritic cells (DCs) and inhibits their apoptosis by enabling them to secrete different cytokines, thereby suppressing T cell immunity and pro-tumorigenic inflammation. In addition, immunosuppressive cells increase the expressions of immune checkpoint proteins, e.g., the PD-1/PD-L1 and CTLA-4, which suppresses the activation of T cells preventing antigen presentation. The expressions of arginase 1 (ARG1), indoleamine 2,3-dioxygenase (IDO), and an inhibitor of differentiation 1 (Id1) can increase the production of ROS and IGF2 and induce an immunosuppressive phenotype. T cell exhaustion and induction of an immunosuppression phenotype can cooperate in promoting tumor growth.

**Table 1 biomedicines-11-00229-t001:** Ligand binding affinities of the three ligands for the different receptor subtypes (EC50, nM).

Receptor Type	Insulin	IGF2	IGF1	Reference
IR-A	0.2–0.9	2.2-9.8	9–41	[4,5,6]
IR-B	0.5–1.6	10–25	30 > 1000	[4,5,6]
IGF1R	30.000–100.000	0.5-4.4	0.2–0.8	[4,7,8]

## Data Availability

Not applicable.

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
