# Peer review of "IGF2: A Role in Metastasis and Tumor Evasion from Immune Surveillance?"

_biomedicines, 2023, doi:10.3390/biomedicines11010229_

Round 1
Reviewer 1 Report
The present review article by Belfiore et al provides a comprehensive overview of IGF2 biology, including physiological and pathological aspects. The article covers important aspects of IGF2 action, such as its role in cancer spread, resistance to therapies and immune-evasion. The paper is very well written and easy to follow. The paper is truly up-to-date and constitutes an extremely important contribution in the area of IGF research. The figures are highly illustrative and didactic. Specific points:
1. Title: the statement ‘IGF2: a role in tumor evasion…’ might be a bit misleading. Evasion from what? This is not clear. Authors should consider adding “…evasion from immune surveillance” or, alternatively “IGF2: a role in tumor progression and metastasis”.
2. Line 48, “…insulin receptor, which comes in two flavors…”. This is a jargon, please replace ‘flavors’ by variants, or molecular forms.
3. Line 82, please close parenthesis after ‘Igf1r’.
4. Line 143, add reference where indicated.
5. All subsections in Section 2 are labeled ‘2.1’.
6. Legend to Figure 1 is divided into three sections (A, B, C). However there are no three panels in this figure. Please, fix.
7. Line 171: authors state ‘SOX2 epigenetically regulates IGF2 expression’. Please, explain this epigenetic mechanism.
8. Lines 186-191: the sentence starting ‘As early of 1995…’, is very long, confusing and grammatically wrong. Please rephrase.
9. Line 210: should be ‘while lung CSCs expressed THEIR cognate receptors’.
10. Line 254: delete ‘of’.
11. Line 466: should be ‘inhibits T lymphocyte proliferation’.
12. The following references include the city and/or full name of the publisher. This information is not necessary and must be removed. References 8, 29, 32, 34, 52, 56, 64, 76, 92, 93, 94, 97, 108, 120, 125, 133.
Author Response
Responses to Reviewers comments.
We wish to thank the Reviewers for the appreciation of our work and for the useful comments and suggestions that have contributed to improve our paper.
Changes in the text of the revised manuscript are highlighted in red.
Comments and Suggestions for Authors
Reviewer 1
The present review article by Belfiore et al provides a comprehensive overview of IGF2 biology, including physiological and pathological aspects. The article covers important aspects of IGF2 action, such as its role in cancer spread, resistance to therapies and immune-evasion. The paper is very well written and easy to follow. The paper is truly up-to-date and constitutes an extremely important contribution in the area of IGF research. The figures are highly illustrative and didactic.
Specific points:
- Title: the statement ‘IGF2: a role in tumor evasion…’ might be a bit misleading. Evasion from what? This is not clear. Authors should consider adding “…evasion from immune surveillance” or, alternatively “IGF2: a role in tumor progression and metastasis”.
According to Reviewer’s suggestion we have modified the title as follows: “IGF2: a role in metastasis and tumor evasion by immune surveillance?”
- Line 48, “…insulin receptor, which comes in two flavors…”. This is a jargon, please replace ‘flavors’ by variants, or molecular forms.
We have replaced “flavors” with “molecular forms”.
- Line 82, please close parenthesis after ‘Igf1r’.
We corrected this error.
- Line 143, add reference where indicated.
Reference has been added.
- All subsections in Section 2 are labeled ‘2.1’.
We have corrected the numbering in the paragraphs of Section 2 and subsequent Sections.
- Legend to Figure 1 is divided into three sections (A, B, C). However there are no three panels in this figure. Please, fix.
We agree with the reviewer: the sections in the figure legend have been removed.
- Line 171: authors state ‘SOX2 epigenetically regulates IGF2 expression’. Please, explain this epigenetic mechanism.
We rephrased the sentence as follows: “Among these transcription factors, SOX2 epigenetically upregulates IGF2 expression, as revealed by strong H3K4me3 signal, a marker of active promoters, at the IGF2 locus of SOX2 expressing cells”. (lines 172-173). The article by Chiu et al. (ref. 57) does not provide much more information on the mechanisms involved.
- Lines 186-191: the sentence starting ‘As early of 1995…’, is very long, confusing and grammatically wrong. Please rephrase.
We thank the reviewer for pointing out this oversight - we have now rephrased the sentence (lines 190-192)
- Line 210: should be ‘while lung CSCs expressed THEIR cognate receptors’.
Corrected.
- Line 254: delete ‘of’.
Corrected.
- Line 466: should be ‘inhibits T lymphocyte proliferation’.
Corrected.
- The following references include the city and/or full name of the publisher. This information is not necessary and must be removed. References 8, 29, 32, 34, 52, 56, 64, 76, 92, 93, 94, 97, 108, 120, 125, 133.
This information has been deleted from all the indicated references.
Reviewer 2 Report
This is a beautifully written review emphasizing the role of IGF2 in tumor evasion and metastasis in the context of tumor micro-environment. The authors emphasized the fact that the patho-physiological role of IGF2 in the tumor micro-environment lags behind that of IGF1 in the literature. Whereas its role in linking high energy intake, increased cell proliferation, and suppression of apoptosis to cancer, are mechanistic studies that have been reported on extensively, little is known about the emerging concept of IGF2, regarding its role in cancer metastasis. In this review, the authors have focused on studies providing compelling evidence that IGF2 plays crucial roles in cancer spread, immune-evasion and therapeutic approaches designed IGF2 action. This is a well written review with excellent presentation and a balanced number of Figures. Only minor suggestions for changes;
1) Line 42 …change ..is depending on the to …depends on the capacity of metastatic foci…..
2) Line 543 Change…Chemotherapy/target therapy to …chemotherapy/targeted therapy
Author Response
Responses to Reviewers’ comments.
We wish to thank the Reviewers for the appreciation of our work and for the useful comments and suggestions that have contributed to improve our paper.
Changes in the text of the revised manuscript are highlighted in red.
Comments and Suggestions for Authors
Reviewer 2
This is a beautifully written review emphasizing the role of IGF2 in tumor evasion and metastasis in the context of tumor micro-environment. The authors emphasized the fact that the patho-physiological role of IGF2 in the tumor micro-environment lags behind that of IGF1 in the literature. Whereas its role in linking high energy intake, increased cell proliferation, and suppression of apoptosis to cancer, are mechanistic studies that have been reported on extensively, little is known about the emerging concept of IGF2, regarding its role in cancer metastasis. In this review, the authors have focused on studies providing compelling evidence that IGF2 plays crucial roles in cancer spread, immune-evasion and therapeutic approaches designed IGF2 action. This is a well written review with excellent presentation and a balanced number of Figures.
Only minor suggestions for changes;
- Line 42 …change ..is depending on the to …depends on the capacity of metastatic foci…..
Corrected.
2) Line 543 Change…Chemotherapy/target therapy to …chemotherapy/targeted therapy
Corrected.